# Otovestibular Symptoms of COVID-19 and Its Vaccines/Treatments

**Marcello Cherchi** [1,2]

1 Ken and Ruth Davee Department of Neurology, Northwestern University Feinberg School of Medicine, Chicago, IL 60611, USA; m-cherchi2@northwestern.edu; Tel.: +1-312-274-0197
2 Chicago Dizziness and Hearing, Chicago, IL 60611, USA

**Definition:** The rapidly developing literature regarding COVID-19 and its treatments has documented an impressive breadth of pathology across multiple organ systems. In this entry, researchers highlight the audiologic and vestibular manifestations that have been reported in association with COVID-19, its vaccines, and some of its treatments.

**Keywords:** COVID-19; vestibular; vertigo; auditory; hearing loss; tinnitus; otovestibular; vaccines

## 1. Introduction

The coronavirus disease pandemic that began in 2019 (COVID-19) poses a range of challenges for patients, physicians, and health care delivery systems. The pandemic's rapidly changing consequences have necessitated commensurately rapid evolution in our understanding of this disease; thus, it is no exaggeration to say that manuscripts on this topic are already out of date even before going to press. The most common presentations (fever, respiratory symptoms, and fatigue) have understandably commanded the most attention, but experience has shown that most organ systems can suffer involvement. Auditory abnormalities and disorders of balance are not among the most common, but accumulating evidence suggests that they are not rare either. In this entry, the author reviews what is known and acknowledge what is not known about the audiologic and vestibular symptoms that have been associated with the disease itself. The author then closes with some comments regarding audiologic and vestibular symptoms reported in association with the vaccines and some of the antiviral treatments for the disease.

## 2. Pathobiology of COVID-19

Transmission of SARS-CoV-2 and entry through the respiratory tract is fairly well understood. Our understanding of how SARS-CoV-2 enters the nervous system is evolving.

### 2.1. How SARS-CoV-2 Infects the Body

The severe acute respiratory syndrome coronavirus type 2 ("SARS-CoV-2") is a member of a family of coronaviruses. This family comprises single-stranded RNA viruses. From their lipid bilayer envelope protrude spike glycoproteins that function as "hooks" that facilitate entry into host cells, where they replicate, and from where they propagate. The spoke glycoproteins bind to angiotensin-converting enzyme receptors [1,2] that are densely expressed in cells in the pulmonary system, which is believed to be the mechanism by which the most common initial symptoms of this infection involve the respiratory tract. However, the infection does not remain confined to the respiratory tract. A more detailed discussion of the molecular aspects underlying this mechanism can be found elsewhere in this Encyclopedia [3].

*2.2. How SARS-CoV-2 Infects the Nervous System*

Investigators have a limited understanding of the factors that make an individual vulnerable to the coronavirus and the mechanisms by which these viruses invade the central nervous system (CNS) [4,5], though there is evidence [6] of cells within the CNS that express angiotensin converting enzyme receptors—even if at a lower density than found in the lower respiratory tract. It appears that after these viruses have invaded the CNS, they are capable of advancing via axonal transport [5].

It has long been known that other coronaviruses can invade the central nervous system, as they have been isolated in the brain and spinal cord tissue or the cerebrospinal fluid of patients suffering from a variety of neurological diseases such as encephalitis [7–11], acute disseminated encephalomyelitis [12], optic neuritis [13], Parkinson's disease [14], and multiple sclerosis [15–17]. For some of these diseases (e.g., multiple sclerosis, Parkinson's disease) it seems somewhat unlikely that coronaviruses play a causative role, whereas in others (e.g., optic neuritis, encephalitis, and encephalomyelitis), it is reasonable to entertain the possibility that the viruses themselves are the etiopathogenetic factor.

Early in the pandemic, clinicians began reporting neurologic symptoms in individuals infected with SARS-CoV-2 [18,19] who had been hospitalized [20], especially among those in intensive care units [21]. Data regarding SARS-CoV-2 [22–24] suggest that its capacity to invade the central nervous system is similar to that of other members of the coronavirus family [25], though there also appear to be clear differences [2], such as the predilection of SARS-CoV-2 to cause anosmia and ageusia [26–30], which in turn suggests involvement of cranial nerves. Therefore it is logical to consider whether cranial nerve involvement by SARS-CoV-2 is the mechanism underlying other focal neurological manifestations, such as the vestibular and auditory symptoms observed in patients presenting to primary care offices, emergency rooms, and otoneurology/neuro-otology clinics.

Early in the pandemic, the author began evaluating patients with laboratory confirmation of SARS-CoV-2 infection who complained of auditory symptoms (such as hearing loss or tinnitus) and/or dizziness. This matches the evolving clinical literature, which reports dizziness [31], auditory symptoms (such as hearing loss or tinnitus), or both [32].

## 3. Auditory and Vestibular Symptoms Associated with COVID-19

The author will briefly review the representative literature regarding COVID-19 and auditory and vestibular symptoms; the author discusses that it is difficult to prove a causal relationship between the infection and these symptoms; and researchers outline several pathogenetic hypotheses regarding the mechanisms that have been proposed in support of the idea that the relationship might be causal.

*3.1. COVID-19 Associated with Auditory Disorders*

The literature is accumulating regarding an association between COVID-19 infection and hearing loss. While a few papers failed to find any relationship between COVID-19 and hearing loss [33,34], a growing number of case reports [35–41] and case series [42–46] have documented clear instances of hearing loss in the context of SARS-CoV-2 infection. The summary article by De Luca et al. [47] reviewed numerous case reports and a case series of COVID-19 patients, including both sexes, with a broad range of ages (18–84 years), with unilateral or bilateral typically sudden onset sensorineural hearing loss. The summary article by Fancello et al. [48] reviewed numerous case reports and two case series of COVID-19 patients, including both sexes, with a similarly broad range of ages, with unilateral (more common) or bilateral (less common) sudden sensorineural hearing loss. The prospective single-institution study by Dusan et al. [42] reported a similar variety.

A smaller but growing literature is also beginning to document an association between COVID-19 and tinnitus [36,37,39,44,45,48–53]. In many cases, the tinnitus accompanies hearing loss [36,37,39]. Some of the literature describes tinnitus occurring with or without concomitant hearing loss [52] or without any reported hearing loss [53].

*3.2. COVID-19 Associated with Disequilibrium*

A case series from Wuhan of 799 patients described dizziness as a symptom in 8% of confirmed COVID-19 patients [54]. Another study, also from Wuhan, of 214 confirmed cases of COVID-19 that specifically asked patients about neurological symptoms reported dizziness in 16.8% of patients [28]. A study of 509 patients from several hospitals in Chicago reported dizziness in 29.7% of cases [20]. The variability reflected in this broad range (8–29.7%) among individual studies has been noted in reviews [31], though it is clear that there is increasing recognition of an association between COVID-19 and vestibular disorders [48]. The ability of COVID-19 to cause auditory and vestibular symptoms appears to be distinct from the other SARS coronaviruses [55].

*3.3. Potential Mechanisms for Auditory and Vestibular Symptoms from SARS-CoV-2*

In view of the emerging evidence of cochlear symptoms (hearing loss, tinnitus) and vestibular symptoms (dizziness) occurring in the context of COVID-19 infection, most reviews of this subject are concluding that the relationship between COVID-19 and audiovestibular dysfunction may be causal rather than simply associative [48,55–65].

In COVID-19 patients with auditory and vestibular symptoms, otovestibular testing is sometimes abnormal [63], though the mechanism by which SARS-CoV-2 causes such dysfunction is unclear. The author discusses several candidate mechanisms here.

First, as mentioned earlier, SARS-CoV-2 seems capable of affecting individual cranial nerves, manifesting with ophthalmoparesis [66], optic neuritis [13], and anosmia [26–30]. It is therefore logical to consider the possibility of SARS-CoV-2 involving the vestibulocochlear nerve, manifesting as vestibular neuritis [67–70] and/or cochlear neuritis or labyrinthitis. It should be noted, however, that some publications [71] describing "COVID-19 vestibular neuritis" appear to base the diagnosis on clinical criteria rather than on objective otovestibular testing; so, this remains a conjecture requiring further study.

Second, there is emerging evidence that the SARS-CoV-2 virus can directly infect inner ear hair cells [72], which may underlie reported cases of labyrinthitis [73]. Whether such involvement by SARS-CoV-2 in turn increases the risk for other specific otologic diseases, such as Ménière's disease [74] and benign paroxysmal positional vertigo [75,76], remains unclear.

A third potential mechanism for inner ear dysfunction due to COVID-19 infection is labyrinthine ischemia secondary to hypoxia [32]. COVID-19 infection increases the risk of thrombosis and can deoxygenate erythrocytes; since the inner ear's circulation is entirely dependent on the small and tenuous labyrinthine artery, this mechanism is plausible.

A fourth potential mechanism is hematologic, via intralabyrinthine hemorrhage, though so far this appears rare [35,77].

A fifth potential mechanism, at least for vestibular (not auditory) symptoms, is that SARS-CoV-2 appears to induce autonomic dysfunction in some patients [78–82]; so, orthostatic intolerance (generally postural orthostatic tachycardia) may be another mechanism by which SARS-CoV-2 provokes dizziness [83–89].

**4. Auditory and Vestibular Symptoms Associated with Vaccination for SARS-CoV-2 and Treatment for COVID-19**

Concerted research and pharmaceutical efforts have been directed at developing vaccines for SARS-CoV-2 and treatments for COVID-19. As these interventions have come to market, new questions have been raised regarding potential adverse effects, including auditory and vestibular symptoms. Here, the author briefly reviews several studies and reports regarding the vaccines and antiviral agents. Other interventions, such as interleukin modulators (anakinra, tocilizumab, and sarilumab) and other immunomodulators being used on an experimental basis in patients who are in (or who are at risk of entering) intensive care units, are covered elsewhere [90].

*4.1. Vaccines for COVID-19*

As of this writing, several vaccines have received FDA approval for use [91–93]. A broader review of possible adverse effects associated with the vaccines can be found elsewhere in this volume [94]. Researchers have received questions from patients regarding whether the vaccines themselves can cause otovestibular symptoms or exacerbate pre-existing otovestibular diseases. There are still insufficient data to answer these questions confidently. Our general experience so far has been that (1) apparent adverse effects from the vaccines have been transient, lasting days to weeks; (2) although symptoms from COVID-19 infection itself can be very mild, they can also be devastating (if a patient survives) or lethal. On the whole, the potential protection conferred by receiving the vaccines appears greatly to outweigh the risk of the vaccines.

4.1.1. Vaccines for COVID-19 and Dizziness

The literature regarding potential adverse effects from the Pfizer vaccine (BNT162b2 mRNA) is somewhat difficult to interpret as it pertains to vestibular symptoms. The original trial [92] included 43,548 participants (of whom 21,720 received the intervention and 21,828 received a placebo). The report, and its supplementary material, do not list dizziness/vertigo as an adverse event. However, a much smaller study (Kadali, Janagama et al. 2021) of 1245 recipients of the vaccine (no placebo arm), documented that out of 803 patients with "generalized" or "neurological" symptoms, 67 (8.34%) reported "dizziness," and 20 (2.49%) reported "vertigo." There has additionally been a case report of postural orthostatic tachycardia occurring following receipt of this vaccine [95]. There is a case report [70] of a patient who received the Pfizer vaccine and subsequently developed vestibular neuritis (supposedly corroborated on video head impulse testing), but whether this relationship was causal or associative is not known.

It is also difficult to interpret the literature about the Moderna vaccine's (mRNA-1273) association with vestibular symptoms. The original trial [91] studied 30,420 participants (of whom 15,210 received the vaccine and 15,210 received a placebo). Symptoms of "dizziness" and "vertigo" were reported as adverse effects of zero patients in the trial's supplementary material. In contrast, a much smaller study of 432 vaccine recipients [96] reported "vertigo like symptoms" in 15 (3.47%) patients and "dizziness" in 63 (14.58%) patients.

It is similarly difficult to interpret the literature about the Johnson & Johnson's Janssen vaccine (Ad26.COV2.S). The original trial [93] included 39,260 participants (of whom 19,630 received the vaccine and 19,630 received placebo). The supplementary material from the trial reported that no vaccine recipients endorsed symptoms of "dizziness" or "vertigo." Yet, the product monograph (https://covid-vaccine.canada.ca/info/pdf/janssen-covid-19-vaccine-pm-en.pdf, accessed 26 June 2022) that reviewed data from 43,783 participants (of whom 21,895 received the vaccine and 21,888 received placebo) stated 13 patients in the treatment group and 7 patients in the placebo group endorsed the symptom of "vertigo."

4.1.2. Vaccines for COVID-19 and Hearing Loss

The Centers for Disease Control's Vaccine Adverse Effects Reporting System (CDC VAERS) provides publicly available data regarding the two mRNA vaccines (Pfizer and Moderna). Analysis of these data revealed 40 cases of unilateral sensorineural hearing loss (confirmed on audiometric testing) thought "most likely" to be attributable to the vaccines themselves [97]. The report indicated that the unilateral hearing loss occurred within 3 weeks (mean 4 days) of having received the vaccines. The calculated incidence was 0.3 cases per 100,000 individuals (i.e., 3 per million).

4.1.3. Vaccines for COVID-19 and Tinnitus

Tinnitus has been less studied as a potential adverse effect of COVID-19 vaccines, but emerging literature is beginning to discuss this [98,99].

*4.2. Anti-Viral Therapies for COVID-19*

For an individual already infected with COVID-19 the vaccines play no role in acute management, but treatments aiming at the virus itself are being investigated. These have recently been reviewed elsewhere [100]. Some of these agents are already known to be ototoxic, such as chloroquine, hydroxychloroquine, and ivermectin.

As of this writing there was considerable interest in the combination antiviral agent Paxlovid, which contains ritonavir (an agent already used in the treatment of HIV) and nirmatrelvir (an investigational agent); unfortunately, there is almost no literature about the potential ototoxicity of these agents; ritonavir is also one component of the combination antiviral drug, Kaletra (which also contains lopinavir), which has been reported to cause reversible bilateral hearing loss [101], but whether that property is due to ritonavir or lopinavir remains unclear.

Bamlanivimab (LY-CoV555 by Lilly) is a neutralizing IgG1 antibody for SARS-CoV-2 [102]. In a randomized phase 3 trial of 1035 patients [103], of whom 518 individuals received bamlanivimab in combination with another antiviral agent (etesevimab), and 517 individuals received a placebo, the symptom of "dizziness" was reported in four patients (0.8%) from the treatment group and three patients (0.6%) from the control group. The comparable rate of this symptom in both groups and the fact that the intervention group received a combination agent (bamlanivimab + etesevimab) makes it unclear what risk of vestibular symptoms bamlanivimab actually incurs.

Molnupiravir (MK-4882/EIDD-2801 by Merck) is a nucleotide analogue that inhibits SARS-CoV-2 replication. In a randomized phase 3 trial completed by 1411 patients [104], of whom 710 individuals received molnupiravir and 701 individuals received placebo, the symptom of "dizziness" was reported in 1% of the treatment group and 0.7% of the placebo group. The comparable rate of this symptom in both groups makes it unclear what risk of vestibular symptoms molnupiravir actually incurs.

**5. Summary**

The accumulating literature suggests that audiologic and vestibular systems can occur in association with COVID-19 infection; such symptoms are not among the most common, nor are they rare. Whether the relationship is causal is difficult to establish, even though some data (such as evidence of viral infection of inner ear hair cells) are quite suggestive of causality. The mechanism through which auditory and vestibular symptoms could be provoked by COVID-19 remains unclear; proposed mechanisms include vestibular (or vestibulocochlear) neuritis; direct infection of cochlear and/or vestibular hair cells; labyrinthine ischemia; intralabyrinthine hemorrhage; and autonomic dysfunction. The pandemic has lasted long enough for investigators to begin to recognize "long-haul" COVID-19 symptoms, often defined as symptoms persisting >6 months beyond initial infection; a recent study of such patients reported that symptoms such as dizziness and tinnitus, if present at the initial evaluation, were either still present or had actually increased by the time of the followup evaluation at 6–9 months [49].

While it is sensible to include COVID-19 on a differential diagnosis during a pandemic, it behooves the clinician to bear in mind that other diseases still merit consideration, and reasonable alternative explanations for a given symptom should be entertained.

As an example, there are emerging reports of patients with diagnosed COVID-19 infection who also appear to have vestibular neuritis. While it may be impossible at this time to ascertain whether the vestibular neuritis is due to COVID-19 itself or has occurred independently, it is still important to identify vestibular hypofunction in this circumstance, because such a deficit is potentially treatable with appropriately targeted vestibular rehabilitation therapy. Thus, in our patients diagnosed with COVID-19 who appear to have an acute vestibular syndrome, researchers recommend undertaking at least a screening otovestibular workup for vestibular weakness, including video head impulse testing, videonystagmography, and ocular and cervical vestibular evoked myogenic potentials; if these results are

compatible with vestibular weakness (whether unilateral or bilateral), then referral for vestibular rehabilitation therapy is logical.

Auditory and vestibular symptoms are also being reported in association with the vaccines and treatments for COVID-19. While causality has not yet been proven, this relationship warrants further scrutiny. However, in view of the evidence of "long-haul" COVID-19 symptoms (mentioned earlier), and the absence (so far) of evidence for similarly lasting symptoms from vaccines and treatments for COVID-19, our current impression is that the overall the risk of harm from the vaccines and treatments seems significantly lower than the risks of SARS-CoV-2 infection itself.

### 6. Conclusions

- The acute, potentially life-threatening manifestations of COVID-19 certainly merit the greatest scrutiny of public health officials, researchers and clinicians. However, once such threats are treated or excluded, the longer-term consequences of COVID-19 are becoming more apparent.
- Among those consequences are auditory and vestibular symptoms associated with COVID-19 itself and with its vaccines and treatments. While these consequences may not be lethal, they still impose significant morbidity which, in turn, can incur non-trivial economic consequences at the individual level and at the broader population-based level.
- Greater understanding of these auditory and vestibular symptoms associated with COVID-19, its vaccines, and its treatments will be achieved partly through continued epidemiologic monitoring.
- Greater understanding of the pathobiological mechanisms underlying these symptoms will require more systematic evaluation of these patients, beginning with more regularly applied audiologic and vestibular testing. Facilitating such evaluations may require changes at the health-care-systems level, perhaps through promulgating practice guidelines.

Accumulating and analyzing such data will improve the likelihood of correctly diagnosing, and perhaps eventually treating, these aspects of the disease.

**Funding:** This research received no external funding.

**Institutional Review Board Statement:** This review article did not require ethical approval from the author's institutional review board.

**Informed Consent Statement:** This study did not involve any patients.

**Data Availability Statement:** This study reviewed literature, and did not collect any new data.

**Conflicts of Interest:** The author declares no conflict of interest.

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
