# Peer review of "Otovestibular Symptoms of COVID-19 and Its Vaccines/Treatments"

_encyclopedia, doi:10.3390/encyclopedia2030080_

Round 1

Reviewer 1 Report

The manuscript is well-written and worth accepting to Encyclopedia.

In this manuscript, Cherchi et al. reviewed the otovestibular symptoms of COVID 19 and its vaccines/treatment.  The manuscript's construction is excellent, and recent advances in this field are well-reviewed.  This manuscript would be helpful for readers to understand the recent topics in this field.

Author Response

Author: Thank you for your comments.

Reviewer 2 Report

This reviewer appreciates the opportunity to evaluate this article with an interesting topic.  The paper is the review that critically reports the relation between the COVID19- vaccine effect and otovestibular symptoms. The manuscript had a valid method, results, and data interpretation and report a topic that is not so specifically reported in the literature. Analysis and discussion are well performed and complete.

Minor revision:  according to the guideline of the journal references should be numbered in square brackets [ ], and placed before the punctuation in the text.

Author Response

Author: Thank you for your comments.  I have re-generated the manuscript through EndNote such that the references are no longer numbered.

Reviewer 3 Report

The topic is very interesting, but frakly speaking there is lack of a few important works:

1. part 4 (line 165), part 4.1 and 4.1.1.:

Please dissuse the issue of side effects in the area of ENT with summary of product characteristics: 

dizziness may occur as a part of product characterists and in Janssen vaccine dissiness and tinnitus is proved

Please cite thi publication: 

https://mdpi-res.com/d_attachment/audiolres/audiolres-12-00025/article_deploy/audiolres-12-00025.pdf?version=1651211160

2. In the seccion anti0-viral treatmet: 

according to the SMPC:

Bamlanivimab -> may cause dizziness

tocilizumab -> may cause dizziness

molnupiravir -> may cause dizziness 

Please discusse it due to this is an infromation form SMPC that is based on clinical trials

Author Response

Author: Thank you very much for bringing to my attention the review article by Skarzynska et al.  I was unaware of that article at the time that I wrote this manuscript, and am happy to be able to reference this very relevant work.  I have incorporated information pertaining to bamlanivimab and molnupiravir.

In the article mentioned by the reviewer, the discussion of tocilizumab appears only to reference information from the drug package insert, rather than a trial or a study, and I was unable to find any data about this in PubMed.  Absent further information, I am hesitant to include this in the discussion of the present manuscript, though I will do so if the editors instruct me so.

Reviewer 4 Report

Otovestibular symptoms of COVID 19 and its vaccines/treatments

The topic is interesting,as the coronavirus disease pandemic that began in 2019 (COVID-19) poses a range of  challenges for patients, for physicians and for health care delivery systems.   

I have only few remarks:

Introduction

However, the infection does not remain confined to the respiratory tract. A more  detailed discussion of the molecular aspects underlying this mechanism can be found else where in this volume (De Masi, Argenio et al. 2022)  Which volume?

In my opinion, the section 3. Auditory and vestibular symptoms associated with COVID-19 should be inserted as second because the author is investigating about pathogenesis of symptoms that are  described  later….

 I think it  coul  be interesting to spend  more words about otovestibular  symptoms….

Literature is accumulating regarding an association between COVID-19 infection and 96 hearing loss Sudden hearing loss? Unilateral, bilateral? Prevalent Age of patients?

Auditory and vestibular symptoms are also being reported in association with the vaccines and treatments for COVID-19. While causality has not yet been proven, this  relationship warrants further scrutiny. Overall the risk of harm from the vaccines and  treatments seem significantly lower than the risks of COVID-19 infection itself.

In my  opinion the discussion about this topic is too short and in this way it can confuse readers.

Why the author decided to present  Otovestibular symptoms of COVID 19 and its vaccines/treat3 ments together? If he thinks they are  related,it  could interesting to discuss about  it

 References

 Should  references be numbered?

Author Response

The topic is interesting,as the coronavirus disease pandemic that began in 2019 (COVID-19) poses a range of  challenges for patients, for physicians and for health care delivery systems.   

Author: Thank you for your comments.

I have only few remarks:

Introduction

However, the infection does not remain confined to the respiratory tract. A more  detailed discussion of the molecular aspects underlying this mechanism can be found else where in this volume (De Masi, Argenio et al. 2022)  Which volume?

Author: Thank you for bringing this to my attention.  I have corrected the data in EndNote; the entry should now read “volume 2”.

In my opinion, the section 3. Auditory and vestibular symptoms associated with COVID-19 should be inserted as second because the author is investigating about pathogenesis of symptoms that are  described  later….

Author: Thank you for this suggestion.  Originally it seemed most logical to me to structure the manuscript by starting with a discussion of the infectious agent (SARS‑CoV-2) and what is known about its pathobiology, and then moving on to a discussion of symptoms related to the disease and to its treatments — though I recognize that this is not the only way of structuring the manuscript.  If the editors would prefer for me to re-order the sections according to this new suggestion, then I will gladly do so.

 I think it  coul  be interesting to spend  more words about otovestibular  symptoms…. 

Literature is accumulating regarding an association between COVID-19 infection and 96 hearing loss Sudden hearing loss? Unilateral, bilateral? Prevalent Age of patients?

Author: Thank you for this suggestion.  I have modified section 3.1 such that the discussion now describes case reports, case series, and reviews, mentioning the diversity of gender, age range, and side(s) of involvement of hearing loss.

Auditory and vestibular symptoms are also being reported in association with the vaccines and treatments for COVID-19. While causality has not yet been proven, this  relationship warrants further scrutiny. Overall the risk of harm from the vaccines and  treatments seem significantly lower than the risks of COVID-19 infection itself. 

In my  opinion the discussion about this topic is too short and in this way it can confuse readers. 

Why the author decided to present  Otovestibular symptoms of COVID 19 and its vaccines/treat3 ments together? If he thinks they are  related,it  could interesting to discuss about  it

Author: Thank you for this suggestion.  I have modified section 5 such that it now includes discussion of “long-haul” COVID, and concludes that while causality remains difficult to prove, the benefit of vaccination/treatment likely outweighs their risks.

 References

 Should  references be numbered?

Author: Thank you for this question.  I have re-generated the manuscript through EndNote such that the references are no longer numbered.

Round 2

Reviewer 2 Report

The author corrected the manuscript according to the suggestions. 

Author Response

Thank you.

Reviewer 3 Report

no comments, the answer for all previous comments was done.

Author Response

Thank you.

Reviewer 4 Report

The topic is interesting, the author answered my questions.

Author Response

Thank you.